# Vitrification of porcine immature oocytes and zygotes results in different levels of DNA damage which reflects developmental competence to the blastocyst stage

**Tamás Somfai**[ID]*, **Seiki Haraguchi, Thanh Quang Dang-Nguyen, Hiroyuki Kaneko, Kazuhiro Kikuchi**

Institute of Agrobiological Sciences, National Agriculture and Food Research Organization, Tsukuba, Ibaraki, Japan

* somfai@affrc.go.jp

**Data Availability Statement:** All relevant data are within the manuscript and its Supporting information files.

## Abstract

The present study investigated the effects of vitrification of porcine oocytes either at the immature Germinal Vesicle (GV) stage before in vitro maturation (GV-stage oocytes) or at the pronuclear stage after in vitro maturation and fertilization (zygotes) on DNA integrity in relevance with their subsequent embryo development. Vitrification at the GV stage but not at the pronuclear stage significantly increased the abundance of double-strand breaks (DSBs) in the DNA measured by the relative fluorescence after γH2AX immunostaining. Treatment of GV-stage oocytes with cryoprotectant agents alone had no effect on DSB levels. When oocytes were vitrified at the GV stage and subjected to in vitro maturation and fertilization (Day 0) and embryo culture, significantly increased DSB levels were detected in subsequent cleavage-stage embryos which were associated with low cell numbers on Day 2, the upregulation of the *RAD51* gene at the 4–8 cell stage (measured by RT-qPCR) and reduced developmental ability to the blastocyst stage when compared with the non-vitrified control. However, total cell numbers and percentages of apoptotic cells (measured by TUNEL) in resultant blastocysts were not different from those of the non-vitrified control. On the other hand, vitrification of zygotes had no effect on DSB levels and the expression of DNA-repair genes in resultant embryos, and their development did not differ from that of the non-vitrified control. These results indicate that during vitrification GV-stage oocytes are more susceptible to DNA damages than zygotes, which affects their subsequent development to the blastocyst stage.

## Introduction

Cryopreservation of oocytes is an important tool for the in vitro preservation of female germ-plasm. In humans, cryopreservation by slow freezing or vitrification of mature oocytes is routinely performed in fertility clinics [1]. Furthermore, vitrification of immature oocytes is

**Funding:** This work (T.S.) was supported by The Japan Society for the Promotion of Science (KAKENHI, grant number: 21K05912). The funders had no role in study design, data collection and analysis, decision to publish, or preparation of the manuscript.

**Competing interests:** The authors have declared that no competing interests exist.

considered to be an alternative approach for the fertility preservation for patients with reproductive disorders, although the results remained unsatisfactory [2]. In pigs, the current spread of African swine fever and other malignant infectious diseases in Asia and Europe emphasizes the importance of in vitro germplasm preservation [3, 4]. It should be noted that porcine oocytes are very sensitive to low temperatures and do not survive traditional slow freezing [5]. On the other hand, high rates of survival (in terms of membrane integrity) have been achieved by the vitrification of porcine oocytes at the immature Germinal Vesicle (GV) stage, mature (metaphase-II) stage, and the pronuclear stage as well [6]. Fertilized porcine oocytes vitrified at the pronuclear stage show little or no reduction in developmental competence to the blastocyst stage and are able to develop to term [7, 8]. Nevertheless, the developmental competence of unfertilized porcine oocytes is severely reduced by the vitrification procedure despite of the high survival rates [6]. Vitrification at the metaphase-II stage simultaneously damages the meiotic spindle and triggers premature parthenogenetic activation in porcine oocytes, which in turn lose their ability to undergo normal fertilization during in vitro fertilization (IVF) and thus fail to cleave and develop to blastocysts [9, 10]. This problem does not arise when oocytes are vitrified at the GV stage [6, 10]. For this reason, in pigs, this stage appears to be suitable for oocyte vitrification. Porcine GV-stage oocytes survived vitrification at high rates by careful adjustments of the parameters for equilibration with cryoprotectant agents (CPA) and for post-vitrification warming [11]. After in vitro maturation (IVM) and IVF some of them could develop to high quality blastocysts and even to term [11]. Nevertheless, vitrification reduces the developmental competence to the blastocyst stage of GV-stage oocytes as well and the exact mechanism(s) behind this phenomenon remained to be elucidated.

Our previous research revealed that vitrification with an optimized protocol does not compromise the ability of GV-stage porcine oocytes to undergo nuclear and cytoplasmic maturation [12] and subsequent fertilization [11, 13]. On the other hand, cryopreservation is known to impair DNA integrity in mammalian oocytes and embryos which could potentially affect development [14]. In mice, oocyte vitrification at the metaphase-II stage was reported to generate double-strand breaks (DSBs) in the DNA even in the subsequently developing 2-cell embryos [15]. In pigs, high frequency of DSBs in early embryos was found to be associated with delayed early development and reduced developmental competence to the blastocyst stage [16]. Delayed development during the early cleavage-stage and reduced developmental competence to the blastocyst stage are indeed typical characteristics of embryos generated from vitrified GV-stage porcine oocytes [17]. Nevertheless, the effects of vitrification on the DNA integrity in immature porcine oocytes and its consequences on subsequent development remained unexplored.

Based on the above mentioned, we hypothesized that vitrification may cause DSBs in the DNA of GV-stage oocytes and subsequently developing embryos which compromises their developmental competence to the blastocyst stage. This hypothesis also suggests lower frequencies of DSBs in vitrified zygotes which showed high developmental competence in earlier studies. The aim of the present study was to clarify the effects of vitrification on DNA integrity in porcine oocytes and zygotes, and to investigate its relevance to subsequent embryo development. First, we assayed the effects vitrification on the frequency of DSBs in DNA of GV-stage oocytes and pronuclear zygotes. Then, the effect of oocyte and zygote vitrification on the frequency of DSBs in DNA of subsequently developing early-stage embryos was investigated in relevance with blastomere numbers. Thereafter, we investigated the effects of oocyte and zygote vitrification on the expression of DNA damage-repair genes in subsequently developing 4-8-cell embryos. Finally, we investigated the effects of oocyte and zygote vitrification on subsequent blastocyst development and the quality of resultant blastocysts measured by total cell numbers and the percentage of apoptotic cells.

## Materials and methods

### Ethics statement

Since live animals were not used in this study, an ethical approval was not required.

### Oocyte collection and IVM

Ovaries of prepubertal cross-bred gilts (Landrace × Large White) were collected at a local
abattoir and transported within 1 h at 35–37˚C to the laboratory in Dulbecco's phosphate-
buffered saline (PBS) (Nissui Pharmaceutical Co., Ltd., Tokyo, Japan). Cumulus-oocyte
complexes (COCs) were collected from 3–6 mm follicles into a collection medium of
Medium 199 (with Hanks' salts; Sigma-Aldrich Corp., St. Louis, MO, USA) supplemented
with 5% fetal bovine serum (FBS, Gibco; Invitrogen Corp., Carlsbad, CA, USA), 20 mM 4-
(2-hydroxyethyl)-1-piperazineethanesulfonic acid (HEPES) (Dojindo Laboratories, Kuma-
moto, Japan) and antibiotics [100 IU/ml penicillin G potassium (Sigma-Aldrich Corp) and
0.1 mg/ml streptomycin sulfate (Sigma-Aldrich)]. The maturation medium was porcine
oocyte medium (POM; Research Institute for the Functional Peptides, Yamagata, Japan)
supplemented with 1 mM dibutyryl cAMP (dbcAMP; Sigma-Aldrich Corp), 10 ng/ml epi-
dermal growth factor (EGF, Sigma-Aldrich Corp), 10 IU/ml eCG (Serotropin; ASKA Phar-
maceutical Co., Ltd., Tokyo, Japan) and 10 IU/ml hCG (Gonatropin; Novartis Animal
Health, Tokyo, Japan). Forty to fifty COCs were cultured in each well of 4-well dishes (Nunc
MultiDishes, Thermo Fisher Scientific) in 500-µl droplets of IVM medium without oil cover-
age for 22 h in an atmosphere of 5% $CO_2$, 5% $O_2$, and 90% $N_2$ at 39˚C as described previously
[11]. Thereafter, COCs were cultured in maturation medium without dbcAMP for an addi-
tional 22–24 h under the same atmosphere.

### IVF and embryo culture

The medium used for IVF was a modified Pig-FM medium [18] containing 10 mM HEPES, 2
mM caffeine, and 5 mg/ml bovine serum albumin (BSA, Sigma-Aldrich). COCs were washed
three times in IVF medium, and transferred into 450-µl IVF droplets (approximately 50
oocytes in each droplet) covered by paraffin oil (Paraffin Liquid; Nacalai Tesque). Frozen-
thawed epididymal spermatozoa from a Meishan boar were preincubated at 37˚C in a sperm
washing medium which was Medium 199 (with Earle's salts, Gibco, pH adjusted to 7.8) sup-
plemented with 4.12 mM calcium lactate, 3.05 mM glucose and 12% (v/v) FBS for 15 min [19].
The sperm pellet was resuspended in 70 µl of the sperm-wash medium, then the concentration
of spermatozoa was adjusted to $1 \times 10^6$ cells/ml by adding IVF medium. To obtain the final
sperm concentration ($1 \times 10^5$ cells/ml), 50 µl of the sperm suspension was introduced into the
IVF droplets containing oocytes and co-incubated for 4 h at 39˚C under 5% $CO_2$, 5% $O_2$ and
90% $N_2$. Thereafter, COCs with attached sperm were transferred to 500-µl aliquots of porcine
zygote medium (PZM)-3 [20] in 4-well culture dishes and subsequently incubated for 4 h.
Then, the COCs were either subjected to vitrification and warming or processed without vitri-
fication, according to the experimental design (detailed below). Thereafter, the cumulus and
spermatozoa were removed from the surface of zona pellucida by gentle pipetting with a fine
glass pipette. The oocytes were then investigated under a stereo microscope to evaluate and
record live/dead status. Those with an intact oolemma, a normal spherical shape, a smooth
surface, and a dark and evenly granular cytoplasm were considered to be alive. The oocytes
with membrane damage and a brownish faded cytoplasm were considered dead and discarded.
Embryo culture was performed in 500 µl drops of porcine zygote medium PZM-3 medium in
4-well culture dishes without oil overlay in an atmosphere of 5% $CO_2$, 5% $O_2$ and 90% $N_2$ at

39˚C. Cleavage was assayed 48 h after IVF under a stereo microscope and total numbers of cleaved embryos and those at or beyond the 4-cell stage were recorded. Blastocyst rates were recorded on Days 5, 6 and 7 (Day 0 = IVF). On Day 5, 10% (v/v) FBS was added directly to culture wells. Numbers of hatching and hatched blastocysts were recorded on Day 7.

## Vitrification/warming of COCs

Vitrification of COCs was performed according to our previous reports [21, 22] either before IVM or after IVF according to the experimental design (detailed below). The basic medium (BM) for vitrification and warming was a modified NCSU-37 [23] without glucose, but supplemented with 20 mM HEPES, 50 μM β-mercaptoethanol, 0.17 mM sodium pyruvate, 2.73 mM sodium lactate and 4 mg/ml BSA. The COCs were briefly washed in BM, pre-warmed to 38˚C. Thereafter, groups of 55–120 COCs were transferred at once to an equilibration medium, composed of BM supplemented with 2% (v/v) ethylene glycol (EG, E-9129), and 2% (v/v) propylene glycol (PG, 29218–35, Nacalai Tesque Inc., Kyoto, Japan). The COCs were incubated in equilibration medium for 13–15 min at room temperature (RT, 25˚C). After equilibration, groups of 25–60 COCs were washed three times in 50-μl of vitrification solution, pipetted into a glass capillary tube, and in 1–2 μL aliquots of vitrification solution dropped onto aluminum foil covered with and floating on liquid nitrogen (LN). The vitrification solution was composed of BM supplemented with 50 mg/mL polyvinylpyrrolidone (P0930, Sigma-Aldrich) 0.3 M sucrose (196–00015, Wako Pure Chemical Industries, Osaka, Japan), 17.5% (v/v) EG and 17.5% (v/v) PG. The COCs were washed in vitrification medium and microdrops of these were placed onto aluminum foil within 30 sec in total. The vitrified droplets were stored until use in 2-ml cryotubes (Iwaki 2732–002; AGC Techno Glass Co. Ltd., Tokyo, Japan) partly immersed in LN. The warming medium was pre-warmed for 3 h to 38.5˚C, and, thereafter, maintained at 42˚C in a dry block tube heater on a warm plate (SP-45D, Hirasawa, Tokyo, Japan) for an additional 20 min. The vitrified droplets were warmed by transfer into 2.5 ml of warming solution (0.4 M sucrose in BM) in a 35 mm plastic dish (Falcon 351008, Becton Dickinson Labware, NJ, USA) on a warm plate that was maintained at 42˚C for immature oocytes [11] or 38.5˚C for zygotes [7]. After 2–3 min, the COCs were transferred for periods of 1 min (each) to 500-μl aliquots of BM supplemented with 0.2, 0.1 or 0.05 M sucrose at 38.0˚C. Thereafter, they were washed in BM without sucrose at 38.0˚C and processed according to the experimental design (detailed below).

## Analysis of fertilization normality after IVF

The fertilization status of oocytes was assessed 10 h after IVF. In each experimental replication, 5–20 oocytes were mounted on glass slides and fixed with acetic alcohol (acetic acid 1: ethanol 3) for at least 5 days, stained with 1% (w/v) orcein (Sigma) in acetic acid, rinsed in glycerol:acetic acid:water (1:1:3) and then examined under a phase-contrast microscope with × 40 or × 100 objectives. The status of oocyte chromatin, the presence and number of female and male pronuclei and/or sperm head(s), and the existence of the first and second polar bodies (1PB and 2PB, respectively) were investigated in the oocytes. The oocytes arrested at the GV stage were omitted from evaluation. The oocytes were considered penetrated when a sperm head(s), and/or male pronucleus(i) with the corresponding sperm tail were detected in the cytoplasm. The oocytes with one penetrating sperm in the cytoplasm were defined as monospermic. Normal fertilization was defined by the presence of 1 female pronucleus and 1 male pronucleus and the extrusion of both 1PB and 2PB. Frequencies of total penetration and normal fertilization were calculated as their percentage from total oocytes. Frequencies of oocytes

with normal male pronuclear formation and monospermy were calculated as their percentage from penetrated oocytes.

## Evaluation of DNA double-strand breaks in oocytes and embryos

DSBs were visualized by immunostaining of the phosphorylated histone H2AX (γH2AX) according to the method of Bohrer et al. [16], with slight modifications. Only live oocytes/ zygotes/embryos selected by morphological appearance were subjected to the assay. Briefly, denuded oocytes, zygotes and cleavage-stage embryos were washed in PBS containing 0.3% (v/ w) polyvinylpyrrolidone (PBS-PVP) then fixed with 4% paraformaldehyde (PFA) (163–20145, Wako Pure Chemical Industries) for 20 min at RT. Then the samples were permeabilized in 1% Triton X-100 in PBS for 30 min at 37˚C. After 3 consecutive washings, the samples were incubated for 1 h at RT in blocking solution (3% BSA and 0.2% Tween-20 in PBS) and then incubated overnight with primary antibody (Anti-phospho-Histone H2A.X (Ser139) antibody, 1:1000 dilution; Merck Millipore, Tokyo, Japan) diluted in blocking solution. Then the samples were washed three times in PBS-PVP supplemented with 0.01% Triton X-100 and 0.1% Tweeen-20 (PBS-TT) and incubated with Alexa Fluor 488–labeled goat anti-mouse IgG (A-10680; Invitrogen) in PBS-TT (1:800) for 1 h at 38.5˚C. Then the samples were washed 3 times in PBS-PVP and once in antifade buffer (S2828, Invitrogen). Then groups of 5–10 oocytes/ embryos were placed on glass slides in 3 μl of a glycerol-based antifade solution (S2828; Invitrogen) supplemented with 0.01 mg/ml Hoechst 33342 (Calbiochem, San Diego, CA, USA) and flattened with a 18×18 mm cover glass. For each oocyte and embryo, images of total DNA and γH2AX labelled with Hoechst 33342 and Alexa Fluor 488, respectively, were taken at the same position with a digital camera (Retiga 2000R Fast1394, QImaging, Canada) under an epi-fluorescent microscope (Olympus BX51, Tokyo, Japan), at the same magnification, focus and camera settings (i.e., gain, exposition) within 5 sec of exposure to UV light using the UV-1A (excitation wavelength of 330–385 nm) and B-2A (excitation wavelength 470–495 nm) filters, respectively. In each replication, samples incubated without the primary antibody were also processed for negative controls, to verify the reliability of assay.

In a previous study, Bohrer et al. [16] counted the numbers of γH2AX foci in the nuclei of cleavage-stage embryos to quantify DSBs after immunostaining. However, our preliminary results showed a generally strong γH2AX signal throughout the DNA of GV-stage oocytes and in some vitrification-related embryos which made it impossible to distinguish single foci (Fig 1). Therefore, to quantify DSB levels in DNA, the fluorescence intensity for γH2AX labelled with Alexa Fluor 488 was measured from digital images using the Image J software (v. 1.52, NIH, Bethesda, MD, USA, https://imagej.nih.gov/ij/). Each image for γH2AX was stacked with its total DNA counterpart. In each image stack, the total area of DNA (labeled with Hoechst 33342) was outlined and within this area the mean signal intensity of Alexa Fluor 488 was measured in the γH2AX-stained image counterpart. Relative DSB levels were calculated from the mean γH2AX fluorescent signal intensities where the control group was considered as 1 (100%).

## Analysis of total cell numbers in blastocysts

To evaluate total cell numbers in embryos, blastocysts on Day 7 (Day 0 = IVF) were fixed in 99.5% ethanol supplemented with 10 μg/mL Hoechst 33342 for an overnight at 4˚C. After washing in ethanol, embryos were mounted on glass slides in glycerol droplets, flattened by cover slips and total nuclei were counted under UV light with excitation at 330–385 nm under an epifluorescent microscope.

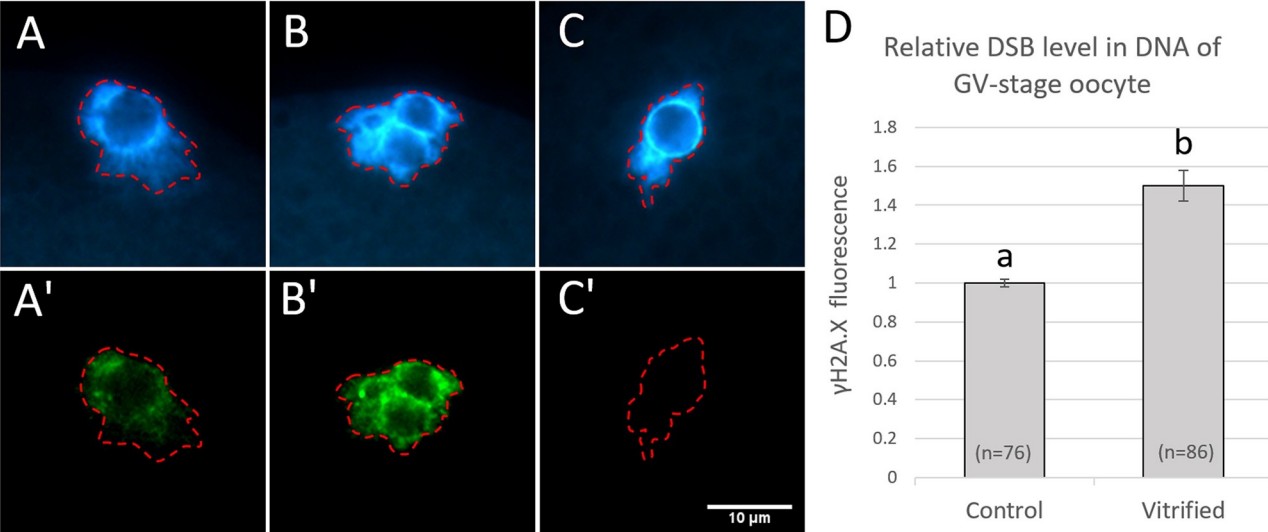

**Fig 1.** Fluorescent of images of DNA (blue, A-C) and the DSB marker γH2AX (green, A'-C') in GV-stage oocytes and relative DSB levels in the DNA of oocytes (D) without and after vitrification and warming. A-A': freshly collected GV-stage oocyte without vitrification; B-B': a freshly collected GV-stage oocyte after vitrification and warming (note increased γH2AX fluorescence and the fragmentation of the nucleolus); C-C': a GV stage oocyte processed without anti-γH2AX (negative control). γH2AX measurement area is highlighted with red dashed line. The scale bare represents 10 μm. Data are shown as mean ± SEM. Values with different superscripts (a and b) differ significantly (P < 0.05). The experiment was replicated five times. Total number of oocytes in each group are given in parentheses.

## Analysis of DNA fragmentation in blastocysts

DNA fragmentation was assessed in blastocyst stage embryos by simultaneous nuclear staining and terminal deoxynucleotidyl transferase (TdT) dUTP nick end labeling (TUNEL) based on the method reported by Karja et al. [24] with modifications. Briefly, the embryos were washed four times in PBS-PVP, and then fixed at room temperature in 4% (w/v) paraformaldehyde diluted in PBS for 1 h. Then, embryos were washed three times in PBS-PVP, permeabilized in 0.1% Triton X-100 (diluted in PBS) for 60 min, and incubated at 4°C overnight in a blocking solution, which was PBS containing 10 mg/mL BSA. Embryos were then washed four times in PBS-PVP and incubated in a TUNEL reaction cocktail (in situ cell death detection kit, Sigma-Aldrich) according to the manufacturer's instructions for 1 h at 38.5°C in the dark. Then, embryos were washed three times in PBS-PVP, and flat-mounted in a glycerol-based anti-fade solution (S2828; Thermo Fisher Scientific) supplemented with 10 μg/ml Hoechst 33342 on glass slides with a cover slip. Labeled nuclei were examined under an epifluorescent microscope (BX-51; Olympus) at excitation wavelengths of 470–495 nm and 330–385 nm for the detection of TUNEL reaction and total DNA (Hoechst 33342), respectively. Cells with DNA labeled by TUNEL were considered be apoptotic. To validate the reliability of the assay, in each experimental replication, 2–5 blastocysts were incubated in 400 IU/mL deoxyribonuclease I (DNase) (D4263; Sigma-Aldrich) for 20 min before the TUNEL reaction, which served as a positive control.

## Quantitative real-time PCR (qRT-PCR) of DNA damage-repair genes

The expression of genes related to DNA damage-repair (Table 1) was analyzed in 4–8 cell embryos using Real-Time PCR. In experimental each replication, pooled samples of 30 embryos were lysed in RLT buffer (RNeasy Micro Kit, QIAGEN, Hilden, Germany) and

**Table 1. Primers used in the present study.**

| Gene | Function | Primer | Access No. |
|------|----------|--------|------------|
| *PGK1* | Glycolysis* | *F*-AGATAACGAACAACCAGAGG | AY677198 |
| | | *R*-TGTCAGGCATAGGGATACC | |
| *RAD51* | HR | *F*-CTTCGGTGGAAGAGGAGAGC | NM_001123181.1 |
| | | *R*-CGGTGTGGAATCCAGCTTCT | |
| *XRCC6* | NHEJ | *F*-AACGGAAGGTGCCCTTTACT | NM_001190185 |
| | | *R*-CTTTTAGCCATTGCCTCAGC | |
| *UDG* | BER | *F*-CAGCTCCGTCAAGAAGATCC | XM_003132925 |
| | | *R*-GCTGAGGTGCTTCTTCCAAC | |
| *MSH2* | MMR | *F*-TGGTCCCAATATGGGAGGTA | NM_001195357 |
| | | *R*-CATTTCAGCCATGAATGTGG | |

*Internal reference gene. HR: Homologous recombination; NHEJ: Non-homologous end-joining; BER: Base excision repair; MMR: Mismatch repair

stored at −80˚C until analysis for each treatment group. Total RNA was extracted and purified from pooled samples using the RNeasy Micro Kit (QIAGEN) according to the manufacturer's instructions. cDNA synthesis was performed using the PrimeScript™ II 1st strand cDNA Synthesis Kit (Takara), following the manufacturer's protocol. The DNA primers of genes investigated in this study are listed in Table 1. Quantitative mRNA analysis of samples was performed using Real-Time PCR at the 1/20 dilution using the LightCycler® 480 SYBR Green I Master (Roche Applied Science, Penzberg, Germany) according to standard protocols. PCR conditions were 95 ºC for 5 minutes and 55 cycles of 95 ºC for 10 seconds, 60 ºC for 5 seconds and 72 ºC for 10 seconds. The quantification of transcripts in samples was analyzed using the LightCycler® 480 Instrument (Roche Applied Science). In each sample, cDNA levels of each gene were calculated directly using the LightCycler® 480 software, comparing measured values to standard curves prepared by the measurement of pooled standards diluted 1 ×, 10 ×, and 100 ×. Relative quantifications of transcripts in each gene were normalized to an internal reference transcript that was *PGK1*, based on the report of Kuijk et al. [25].

## Experimental design

**Experiment 1. The effect of immature oocyte vitrification and zygote vitrification on DSB levels in DNA.** COCs collected from the same lot of ovaries were either vitrified and warmed at the GV stage immediately after collection (GV-vitrified group) or subjected to IVM and IVF and vitrified/warmed 8 h after IVF (zygote-vitrified group). Relative DSB levels in DNA of vitrified oocytes and zygotes were measured after warming as described above and compared to those of non-vitrified (control) oocytes and zygotes sampled at the same timepoint. Five biological replications were performed.

**Experiment 2. The effect of CPA treatment on DSB levels in the DNA of immature oocytes.** Since Experiment 1 revealed elevated DSB levels in vitrified GV-stage oocytes (but not in vitrified zygotes), Experiment 2 was performed to clarify if this phenomenon occurs due to CPA treatment. Immediately after collection, the COCs were treated with CPA and warming solutions without vitrification as described above (CPA group). Relative DSB levels in the DNA of oocytes were measured as described above and compared to those of non-treated (control) oocytes sampled at the same timepoint. Five biological replications were performed.

**Experiment 3. The effect of immature oocyte vitrification and zygote vitrification on DSB levels in the DNA of subsequently developing cleavage-stage embryos.** COCs collected from the same ovary lot were either vitrified and warmed at the GV stage immediately after collection (GV-vitrified group) and subjected to IVM and IVF or subjected to IVM and IVF and vitrified/warmed at 8 h after IVF (zygote-vitrified group) or subjected to IVM and IVF without vitrification at any stage (control). After IVF, live oocytes were selected as described above and subsequently cultured. Survival rates (expressed as the living oocytes on the total number of oocytes) were recorded. Total numbers of cleaved embryos and those at or beyond the 4-cell stage were recorded 48 h after IVF. All cleaved embryos were subjected to simultaneous labeling of DNA and γH2AX as described above. Total cell numbers in embryos and relative DSB levels in their nuclei were compared among the groups. Five biological replications were performed.

**Experiment 4. The effect of immature oocyte vitrification and zygote vitrification on the expression of DNA damage-repair genes in subsequently developing embryos.** GV-vitrified, zygote-vitrified and control groups were set up as described in Experiment 3. After IVF, survival rates were recorded as described above and live oocytes were subsequently cultured. Total numbers of cleaved embryos and those at or beyond the 4-cell stage were recorded 48 h after IVF. In each group, embryos at the 4–8 cell stage were harvested 55 h after IVF to compare the expression of DNA damage-repair genes by qRT-PCR as described above. Three biological replications were performed.

**Experiment 5. The effect of immature oocyte vitrification and zygote vitrification on fertilization, embryo development to the blastocyst stage, and the quality of blastocysts.** GV-vitrified, zygote-vitrified and control groups were set up as described in Experiment 3. After IVF, survival rates were recorded as described above and live oocytes were subsequently cultured. Total numbers of cleaved embryos and those at or beyond the 4-cell stage were recorded 48 h after IVF. The embryos were further cultured. Blastocyst development rates of cultured oocytes on Day 5, 6 and 7 (Day 0 = IVF) and the combined rates of hatching+hatched blastocysts on Day 7 were compared among groups. On Day 7, total cell numbers and the percentage of apoptotic cells in blastocysts were compared among the groups. Eight biological replications were performed. In 6 replications, fertilization normality was assayed in representative presumptive zygotes in each group as described above.

## Statistical analysis

Data are expressed as the mean ± SEM. All data were analyzed by the GraphPad Prism software (Version 7.02 for Windows, GraphPad Software, La Jolla, California, USA). In experiments where pairs of groups were compared (*Experiment 1* and *2*), Mann-Whitney unpaired t-test was applied according to software protocol since normal distribution was not verified. Overall survival and cleavage rates of Experiment 3, 4 and 5 (employing the same treatment groups) were summarized for a total of 16 replications. Percentage data were arcsine transformed and normal distribution was verified by the D'Agostino and Pearson normality test. Pearson correlation coefficient was calculated to assay the relationship between the numbers of vitrified oocytes or zygotes in each group and the percentage of survival. Thereafter percentages of survival and cleavage were analyzed by one-way ANOVA followed by Tukey's multiple comparisons test. Results of blastocyst development in *Experiment 5* (showing normal distribution) were analyzed the same way. In the rest of experiments where more than 2 groups were compared but normal distribution could not be verified, percentage data were arcsine transformed and analyzed by the Kruskal-Wallis test followed by Dunn's multiple comparison, according to software protocol. $P < 0.05$ was defined as the significance level.

## Results

### The effect of immature oocyte vitrification and zygote vitrification on DSB levels in DNA

When COCs were vitrified at the GV stage, the fluorescence intensity of γH2AX and thus the relative level of DSBs in DNA were significantly increased after warming compared with their fresh (non-vitrified) counterparts (Fig 1). When COCs were vitrified and warmed after IVM, IVF and 4 h of subsequent culture, the oocytes had pronuclei and the relative level of DSBs in them did not differ significantly from that of their non-vitrified counterparts (Fig 2). Both in GV-stage oocytes and pronuclear zygotes, fluorescent signals for γH2AX were detected only within the DNA and signals were lacking in negative control oocytes (Figs 1 and 2).

### The effect of CPA treatment on DSB levels in DNA of GV-stage oocytes

When immature COCs were treated sequentially with equilibration, vitrification and warming solutions, the mean level of relative level of DSBs in DNA was not significantly different from that of the untreated control (S1 Fig).

### The effect of immature oocyte vitrification and zygote vitrification on DSB levels in the DNA of subsequently developing cleavage-stage embryos

Vitrification of COCs at the GV-stage significantly increased the fluorescence intensity of γH2AX and thus the relative level of DSBs in DNA in subsequently obtained cleavage-stage embryos 48 h after IVF compared with the non-vitrified control and this was associated with reduced cell numbers (Fig 3). When zygotes were vitrified and warmed at the pronuclear stage, the relative level of DSBs in DNA and the mean number of blastomeres in subsequently obtained embryos did not differ significantly from that of the non-vitrified control. In all

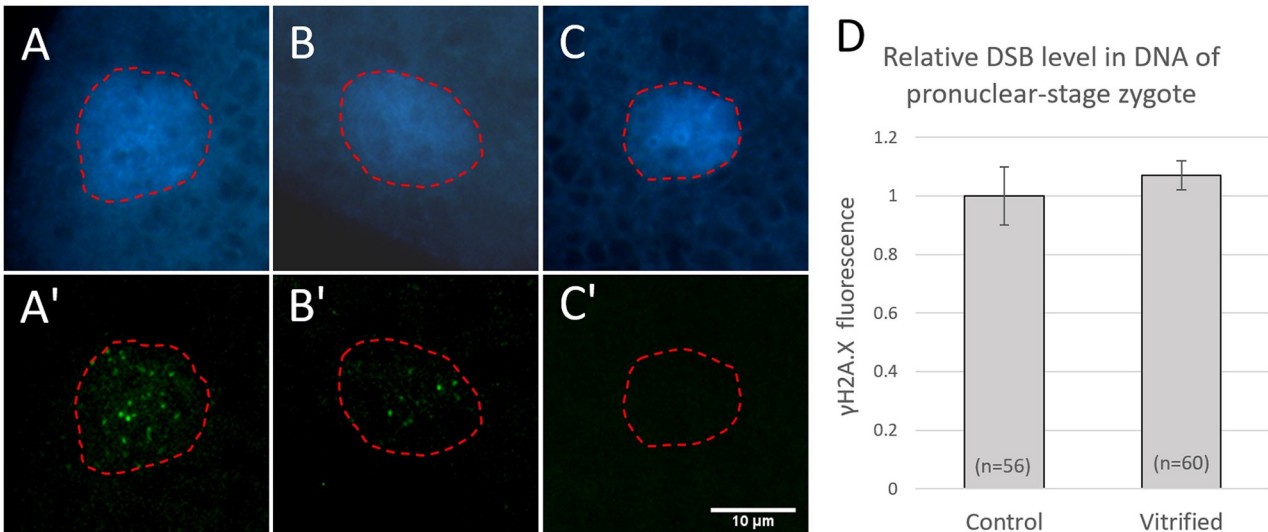

**Fig 2.** Fluorescent images of DNA (blue, A-C) and the DSB marker γH2AX (green, A'-C') in pronuclei of zygotes 8 h after IVF and relative DSB levels (D) without and after vitrification and warming. A-A': a representative pronucleus in a zygote without vitrification; B-B': a representative pronucleus in a zygote after vitrification and warming; C-C': a representative pronucleus in a zygote processed without anti-γH2AX (negative control). γH2AX measurement area is highlighted with red dashed line. The scale bare represents 10 μm. Data are shown as mean ± SEM. Significant difference was not observed in DSB levels between the groups (P ≥ 0.05). The experiment was replicated five times. Total number of zygotes in each group are given in parentheses.

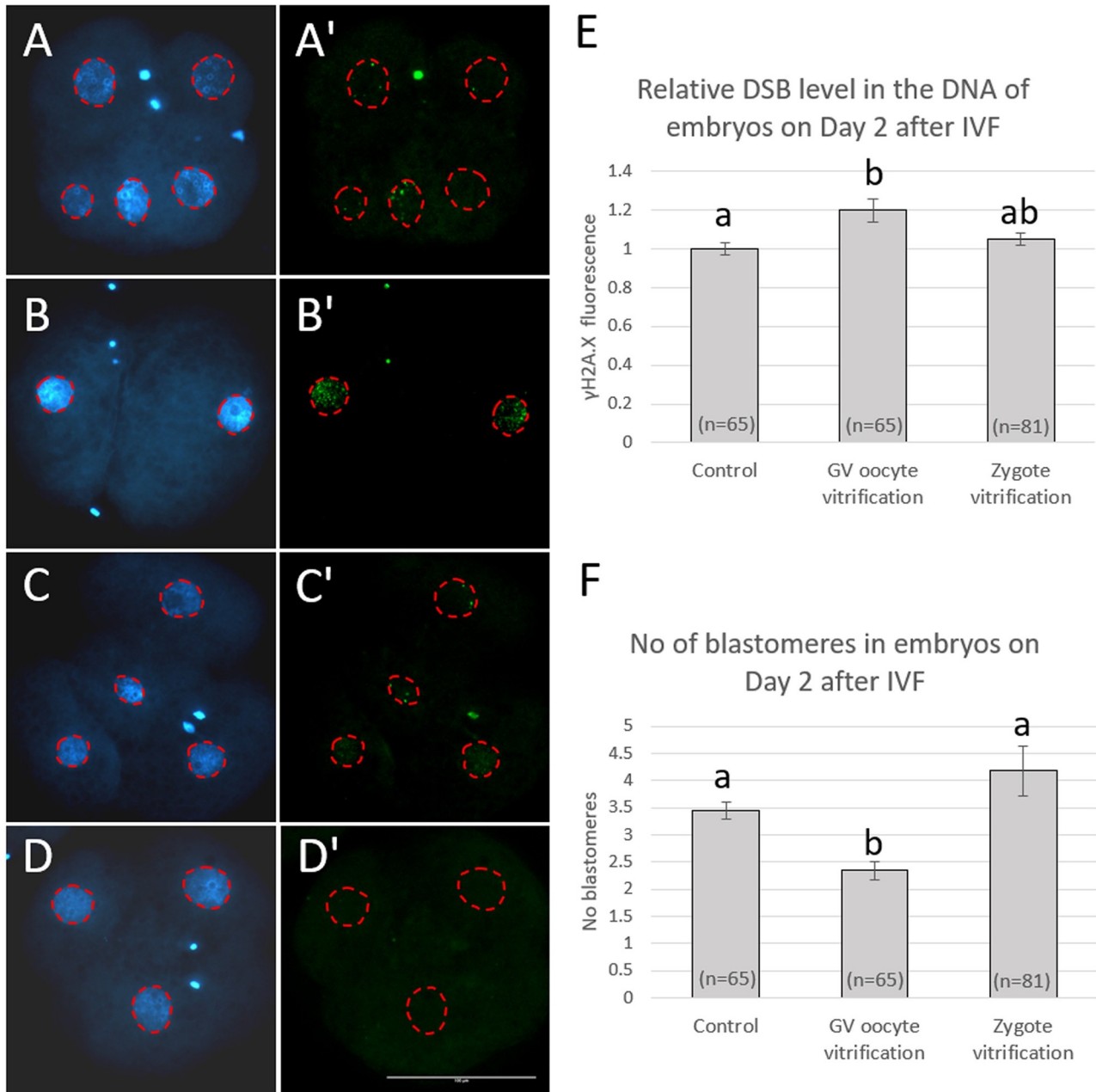

**Fig 3.** Fluorescent images of DNA (blue, A-D) and the DSB marker γH2AX (green A'-D') in cleavage-stage embryos 48 h after IVF, relative DSB levels (E) in embryos obtained from oocytes without vitrification (Control), oocytes vitrified/warmed at the GV-stage and pronuclear zygotes vitrified 8 h after IVF, and (F) numbers of blastomeres in respective embryos. A-A': a representative embryo obtained without vitrification; B-B': a representative embryo obtained from a vitrified GV-stage oocyte; C-C': a representative embryo obtained from a vitrified zygote; D-D': a representative embryo processed without anti-γH2AX (negative control). γH2AX measurement area is highlighted with red dashed line. The scale bare represents 100 μm. Data are shown as mean ± SEM. Values with different superscripts (a and b) differ significantly (P < 0.05). The experiment was replicated five times. Total number of oocytes/embryos in each group are given in parentheses.

**Table 2. Survival and cleavage after IVF of cumulus-enclosed porcine oocytes vitrified either at the GV stage before IVM (GV-vitrified) or at the zygote stage after IVM/IVF (Zyg-vitrified).**

| Group | Total | Survived (% total) | Cultured | Cleaved* (% cultured) | At or beyond the 4-cell stage* (% cleaved) |
|---|---|---|---|---|---|
| **Control** | 751 | 732 (97.3±0.2)[a] | 597 | 391 (64.8±7.7)[a] | 279 (69.5±3.8)[a] |
| **GV-vitrified** | 1327 | 1156 (87.6±1.1)[b] | 1044 | 452 (40.9±3.1)[b] | 223 (45.5±3.6)[b] |
| **Zyg-vitrified** | 962 | 917 (95.3±1.0)[a] | 823 | 492 (59.0±4.3)[a] | 380 (75.6±4.2)[a] |

Sixteen replications were performed. Percentage data are shown as mean ± SEM. Different superscripts in the same column (a,b) denote significant differences (P<0.05).

*Recorded on Day 2 (IVF = Day 0)

treatment groups, fluorescent signals for γH2AX were detected only within the DNA and signals were completely lacking in the negative control group (Fig 3).

## The effect of immature oocyte vitrification and zygote vitrification on overall survival, fertilization, and early developmental progress

In Experiments 3, 4 and 5, the survival rates were recorded after IVF in each group in a total of 16 replications. Furthermore, cleavage rates, and numbers of blastomeres in cleaved embryos were recorded on Day 2 in representative groups of embryos (Table 2). The number of oocytes or zygotes vitrified in a single group had no effect on survival rates (S2 Fig). Overall survival and cleavage rates of oocytes in the GV-vitrified group were significantly lower than those of the non-vitrified control group and the zygote-vitrified group (87.6% vs 97.3% and 95.3%, and 40.9% vs 64.8% and 59.0%, respectively) (Table 2). Furthermore, the percentage of cleaved embryos at-or-beyond the 4-cell stage at 48 h after IVF in the GV-vitrified group was significantly lower than those in the control and zygote-vitrified groups (45.5% vs 69.5% and 75.6%, respectively). There was no significant difference in the percentages of survival, cleavage and cleaved embryos at-or-beyond the 4-cell stage at 48 h after IVF between the control and the zygote-vitrified groups (Table 2).

## The effect of immature oocyte vitrification and zygote vitrification on the expression of DNA damage-repair genes in subsequently developing embryos

The relative expression of *RAD51* in 4–8 cell embryos in the GV-vitrified group was significantly increased compared with the control and zygote-vitrified groups (Fig 4). There was no statistical difference between the 3 treatment groups in the expression levels of *XRCC6*, *UDG* and *MSH2* genes in 4–8 cell embryos (Fig 4).

## The effect of immature oocyte vitrification and zygote vitrification on fertilization, subsequent embryo development to the blastocyst stage, and the quality of blastocysts

The percentages of sperm penetration, monospermy, male pronuclear formation and normal fertilization pattern (i.e. oocytes having 2 polar bodies and 2 pronuclei) were similar among the control, GV-vitrified and zygote-vitrified groups (S1 Table). In vitro development to the blastocyst stage of presumptive zygotes obtained in the three treatment groups is shown in Table 3. From Day 5 to Day 7, the percentage of blastocyst for formation in the GV-vitrified group was significantly lower than those in the control and the zygote-vitrified groups, which in turn did not differ significantly from one another (Table 3).

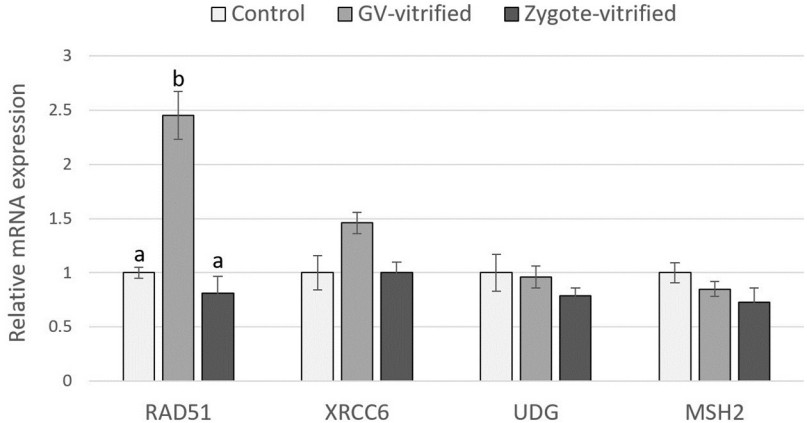

**Fig 4. Relative mRNA expression of DNA-repair genes in 4–8 cell stage embryos obtained either without vitrification (Control) or after oocyte vitrification at the GV-stage (GV-vitrified) or after zygote vitrification (Zygote-vitrified).** Data are shown as the mean ± SEM. Values with different superscripts (a and b) among treatment groups differ significantly (P < 0.05). The experiment was replicated three times.

In a similar manner, the percentage of blastocysts showing the signs of hatching on Day 7 in the GV-vitrified group was significantly reduced in the GV-vitrified group, compared with the control and the zygote-vitrified groups which were in turn similar to one another (Table 3). There was no significant difference between the three treatment groups in terms of the numbers of total blastomeres and the percentages of apoptotic cells in blastocysts (Table 4, Fig 5).

**Table 3. In vitro development to the blastocyst stage of porcine oocytes vitrified either at the GV stage (GV-vitrified) or at the zygote stage (Zyg-vitrified).**

| Treatment groups | Total cultured* | Blastocysts | | | |
|---|---|---|---|---|---|
| | | Day 5 (% cultured) | Day 6 (% cultured) | Day 7 (% cultured) | Hatching (% Day 7 blastocysts) |
| **Control** | 304 | 65 (21.0±2.2)[a] | 105 (33.7±3.9)[a] | 109 (34.0±3.9)[a] | 29 (24.3±5.4)[ab] |
| **GV-vitrified** | 539 | 19 (3.4±1.7)[b] | 49 (8.4±2.0)[b] | 58 (10.2±2.2)[b] | 5 (4.9±2.9)[b] |
| **Zyg-vitrified** | 410 | 82 (19.2±5.6)[a] | 130 (30.5±6.8)[a] | 138 (32.5±6.5)[a] | 50 (28.0±6.5)[a] |

Eight replications were performed. Percentage data are presented as mean ± SEM. Percentage data are shown as mean ± SEM. Different superscripts in the same column (a,b) denote significant differences (P<0.05).

* Only live oocytes were cultured.

**Table 4. Total cell numbers and apoptosis in Day 7 blastocyst obtained from oocytes vitrified either at the GV stage (GV-vitrified) or at the zygote stage (Zyg-vitrified).**

| Group | Total blastocysts analyzed | Total cells | Analyzed by TUNEL* | % Apoptotic cells |
|---|---|---|---|---|
| **Control** | 109 | 71.3±5.8 | 53 | 11.6±0.1 |
| **GV-vitrified** | 58 | 65.0±10.3 | 37 | 7.0±0.9 |
| **Zyg-vitrified** | 138 | 79.7±6.1 | 69 | 7.3±1.8 |

Eight replications were performed.

*Analyzed in 3 replications. Percentage data are shown as mean ± SEM.

Significant differences were not detected among groups.

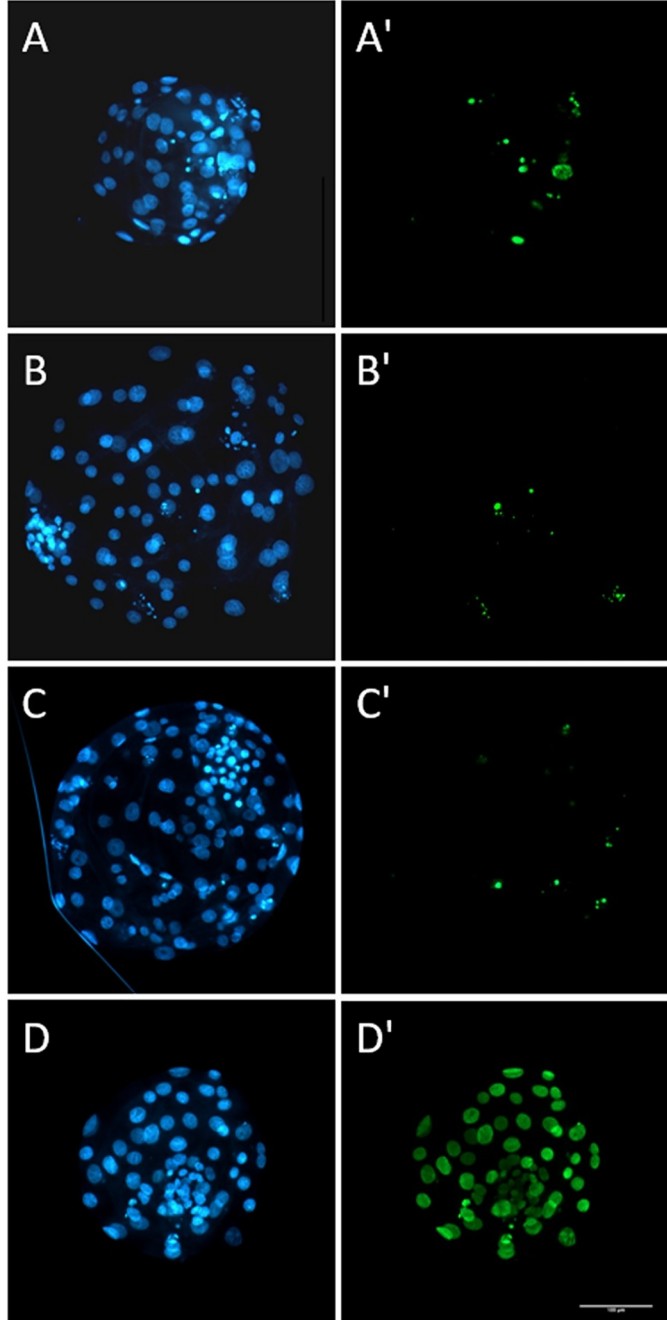

**Fig 5.** Fluorescent images of total nuclei (blue, A-D) and TUNEL positive nuclei (green, A'-D') in blastocysts on Day 7. A-A': a representative embryo obtained without vitrification; B-B': a representative embryo obtained from a vitrified GV-stage oocyte; C-C': a representative embryo obtained from a vitrified zygote; D-D': a representative embryo processed after DNase treatment (positive control). The scale bare represents 100 μm.

## Discussion

In the present study, we assayed the effects of vitrification on relative DSB (γH2AX) levels in the DNA of porcine immature oocytes at the GV stage and pronuclear zygotes after warming and also in subsequently developing cleavage-stage embryos. Furthermore, we investigated the

effects of oocyte and zygote vitrification on subsequent in vitro development to the blastocyst stage and the expression of DNA-damage-repair genes in embryos. In several previous studies, our team has investigated the developmental competence to the blastocyst stage of porcine oocytes vitrified at the immature oocyte stage [10, 11, 13, 21, 26] or zygotes vitrified at the pronuclear stage [7, 8]. However, this is the first study to directly compare the effects of vitrification at these stages on embryo development in relevance with DNA integrity.

In the present study, 95.3% of the presumptive zygotes survived vitrification which was not significantly different from the percentage of live oocytes in the control group. Although the mean survival rate of GV-stage oocytes was significantly lower, it was still a relatively high 87.6%. Nevertheless, after vitrification at the GV stage, surviving oocytes showed reduced competence for cleavage and subsequent embryo development unlike zygotes vitrified by the same protocol. These results are in agreement with our previous reports in pigs [8, 11, 21, 27] and cattle [22, 28]. Previously we demonstrated that our vitrification protocol does not impair the ability of GV-stage oocytes to undergo maturation and subsequent fertilization [11–13, 26]. Orcein staining of oocytes after IVF in the present study also confirmed that vitrification of oocytes at the GV stage or at the zygote stage had no detrimental effect on fertilization and subsequent pronuclear formation. This suggests that a factor(s) other than meiotic or fertilization malfunction caused the reduced ability of vitrified oocytes to undergo cleavage and subsequent embryo development.

Our current results reveal that vitrification of COCs at the GV stage caused DSBs in the DNA of the oocyte whereas vitrification at the zygote stage did not affect the number of DSBs in the DNA within the pronuclei (Experiment 1). Cryoprotectants used in this study such as ethylene glycol and especially propylene glycol are known to generate DSBs in oocytes [29, 30]. However, the CPA treatment itself was not responsible for the increase of DSBs in GV-stage oocytes (Experiment 2) which corresponds with our previous results indicating that the CPA treatment protocol itself did not reduce the developmental competence of immature porcine oocytes [26, 31]. This suggests that during vitrification and warming, DSBs could have been caused by the extreme temperature changes or the combined effects of temperature changes and CPA. Another remarkable finding of the present study is that a certain amount of DSBs caused by vitrification at the GV-stage remained in the DNA of the oocyte and the resultant embryo during subsequent IVM, IVF and embryo culture at least up to Day 2 which was associated with reduced number of blastomeres in cleaved embryos at this timepoint (Experiment 3). These results are in accordance with our previous study reporting reduced blastomere numbers in Day 2 in embryos obtained from vitrified immature oocytes [17]. In the present study, 45.5% of the cleaved embryos in the GV-vitrified group were still at the 2-cell stage at 48 h after IVF, whereas both in the control and zygote-vitrified groups most of the embryos (69.5–75.6%) already reached the 4-cell stage by this timepoint. The relationship between high DSBs numbers in nuclear DNA and compromised/delayed early development in cleavage-stage porcine embryos has been reported previously [16], which suggests that increased numbers of DSBs the embryos obtained from vitrified GV-stage oocytes may be responsible for their compromised development. It is not clear if embryos in this group showing low cell numbers (i.e. 2–3 cells) at 48 h after IVF remain permanently arrested at this stage or still capable to develop forward and form blastocysts. Unrepaired DSBs can cause permanent developmental arrest and death of eukaryotic cells [32, 33]. Furthermore, in porcine embryos, maternal to zygotic genome activation (de novo mRNA transcription) starts at the 4-cell stage [34]. This is a crucial step for embryonic development and porcine embryos often react to suboptimal environmental conditions (stress) with permanent developmental arrest at the 4-cell stage [35]. Therefore, is plausible that many of the embryos obtained from vitrified oocytes remain permanently arrested at the 2–4 cell stage. On the other hand, vitrification of porcine oocytes at

the GV-stage is known to delay the formation of blastocysts during subsequent in vitro embryo production [11]. Therefore, it is also possible that in some resultant embryos, early development is only delayed but not permanently arrested and such embryos can proceed development to later stages. Further research using time lapse cinematography will be needed to clarify this point.

In mammals, homologous recombination (HR) and non-homologous end-joining (NHEJ) are generally considered to be the main pathways for DSB repair [36, 37]. Accordingly, the increase of DSBs in porcine embryos was associated with the upregulation of representative genes for HR and NHEJ such as *RAD51* and *XRCC6*, respectively [16, 38]. On the other hand, genes for other DNA repair pathways such as base excision repair (BER) and mismatch repair (MMR) are also expressed in porcine oocytes [39]. To have an insight into the effect of oocyte/zygote vitrification on the DNA damage-repair mechanisms in resultant embryos we analyzed the expression of representative genes for HR (*RAD51*), NHEJ (*XRCC6*), BER (*UDG*) and MMR (*MSH2*) in 4–8 cell stage embryos obtained from vitrified GV-stage oocytes, vitrified zygotes, and non-vitrified oocytes (Experiment 4). The expression levels of these genes were similar between the control and zygote-vitrified groups. However, in the GV-vitrified group, *RAD51* was found to be upregulated. In eukaryotes, *RAD51* plays an important role in DNA repair by HR [40]. Upregulation of the *RAD51* indicates an increased activity of DNA repair in embryos obtained from vitrified GV-stage oocytes, which was not observed in embryos obtained from vitrified zygotes. This result corresponds with the findings of Experiment 1 and 3 and supports the conclusions that 1) during vitrification GV-stage immature oocytes suffered a greater level of DNA damage than did pronuclear stage zygotes (which in turn were similar to the control) and that 2) to some extent, DSBs generated by vitrification at the GV stage are carried through oocyte maturation, fertilization and early embryo formation and potentially affect the development of the resultant embryo. The result that only *RAD51* but not the *XRCC6*, *UDG* and *MSH2* genes were upregulated in embryos of the GV-vitrified group is in accordance with previous findings reported by Bohrer et al. [36] that HR is the main pathway to repair DSBs in early porcine embryos.

In porcine oocytes and embryos, *RAD51* has a key importance for normal development by maintaining chromosome integrity and mitochondrial functions [37, 41]. In porcine embryos, the inhibition of *RAD51* increased the number of the γH2AX foci (DSBs) and also induced apoptosis [37]. In the present study, increased levels of DSBs were detected in the cleavage-stage (2–4 cell stage) embryos obtained from vitrified GV-stage oocytes, and this was associated with an increased *RAD51* expression at the 4–8 cell stage. However, the quality of subsequently developing blastocysts in terms of total cell number and apoptosis was not reduced, in accordance with our previous studies [11, 13, 26]. It is possible that, after oocyte vitrification, upregulation of *RAD51* at and beyond the 4-cell stage is important for further DSB repair in resultant embryos preventing apoptosis and ensuring normal cell numbers in resultant blastocysts. Further research will be necessary to clarify this point.

Taken together, the present results reveal a higher tolerance of pronuclear zygotes to vitrification-induced DNA damage compared with immature oocytes. The exact reason for this phenomenon is yet to be clarified. Differences between immature oocytes and zygotes in the permeability of their membranes to water and CPA could theoretically cause different levels of DNA damage. The membrane of mammalian oocytes before maturation is less permeable by CPA and more permeable by water than that after maturation [42, 43]. This may cause an insufficient CPA inflow and a faster increase in intracellular solute concentration when GV-stage oocytes are placed in the highly osmotic vitrification solution making them more susceptible to damages caused by temperature changes and osmotic stress compared with zygotes. Fragmentation of the nucleolus was frequently observed in vitrified oocytes in the present

study (Fig 1B) as well as in our previous report [12] which suggests considerable mechanical forces upon the GV during the process vitrification/warming caused by hyperosmocity. Hyperosmocity can induce DBS in the DNA via various pathways [14]. Nevertheless, this might not be the main reason in vitrified GV-stage oocytes since CPA treatment alone did not increase DSB levels in Experiment 2. Further research will be necessary to clarify this point.

In conclusion, the present study revealed that vitrification and warming at the GV stage but not at the pronuclear stage significantly increased the level of DSBs in the DNA. This was not caused by CPA treatment. Furthermore, after oocyte vitrification at the GV stage, increased DSB levels remained in subsequently developing cleavage-stage embryos which was associated with low cell numbers on Day 2, reduced development to the blastocyst stage and the upregulation of the *RAD51* gene at the 4–8 cell stage. Nevertheless, cell numbers and apoptosis in resultant blastocysts were normal. On the other hand, vitrification of pronuclear zygotes had no effect on DSB levels and the expression of DNA-repair genes in resultant embryos, and their development did not differ from that of the non-vitrified control. These results indicate that GV-stage oocytes are more susceptible to DNA damages caused by vitrification than pronuclear zygotes, which potentially affects their subsequent development to the blastocyst stage.

## Supporting information

**S1 Fig. Relative DSB levels in the DNA of GV-stage oocytes without (Control) and after treatment with CPA and warming solutions (CPA) used for vitrification in this study.** Data are shown as the mean ± SEM. A significant difference was not detected (P > 0.05) between the groups. The experiment was replicated three times. Total numbers of oocytes in each group are given in parentheses.
(TIF)

**S2 Fig. Correlation between the total numbers of oocytes or zygotes vitrified in groups and the percentages of survival.** A significant correlation was not detected (P > 0.05).
(TIF)

**S1 Table. Fertilization status 10 h after IVF of porcine oocytes vitrified either at the GV stage (GV-vitrified) or at the zygote stage (Zyg-vitrified).** Six replications were performed. Percentage data are presented as mean ± SEM. No significant differences were detected among the treatment groups (P < 0.05).* Only live oocytes were used to assess fertilization status. **Characterized by one female pronucleus, one male pronucleus and 2 polar bodies.
MPN = male pronucleus.
(DOCX)

**S1 Data.**
(XLSX)

## Acknowledgments

We thank Ms. Mitsuru Nagai, for technical assistance.

## Author Contributions

**Conceptualization:** Tamás Somfai, Seiki Haraguchi.

**Data curation:** Tamás Somfai.

**Funding acquisition:** Tamás Somfai.

**Investigation:** Tamás Somfai.

**Resources:** Kazuhiro Kikuchi.

**Supervision:** Kazuhiro Kikuchi.

**Writing – original draft:** Tamás Somfai.

**Writing – review & editing:** Seiki Haraguchi, Thanh Quang Dang-Nguyen, Hiroyuki Kaneko.

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
