## [Decision Letter · Decision Letter 0]

24 Jan 2023

PONE-D-22-34298Vitrification of porcine immature oocytes and zygotes results in different levels of DNA damage which reflects developmental competence to the blastocyst stagePLOS ONE

Dear Dr. Somfai,

Thank you for submitting your manuscript to PLOS ONE. After careful consideration, we feel that it has merit but does not fully meet PLOS ONE’s publication criteria as it currently stands. Therefore, we invite you to submit a revised version of the manuscript that addresses the points raised during the review process.

We look forward to receiving your revised manuscript.

Kind regards,

Shao-Chen Sun

Academic Editor

PLOS ONE

Journal Requirements:

"We thank Ms. Mitsuru Nagai, for technical assistance. This work was supported by JSPS KAKENHI (Grant number: 21K05912)."

"This work (T.S.) was supported by The Japan Society for the Promotion of Science (KAKENHI, grant number: 21K05912).

Reviewers' comments:

Reviewer's Responses to Questions

**Comments to the Author**

1. Is the manuscript technically sound, and do the data support the conclusions?

Reviewer #1: Yes

Reviewer #2: Yes

Reviewer #3: Yes

2. Has the statistical analysis been performed appropriately and rigorously? 

Reviewer #1: Yes

Reviewer #2: Yes

Reviewer #3: Yes

3. Have the authors made all data underlying the findings in their manuscript fully available?

Reviewer #1: Yes

Reviewer #2: Yes

Reviewer #3: Yes

4. Is the manuscript presented in an intelligible fashion and written in standard English?

Reviewer #1: Yes

Reviewer #2: Yes

Reviewer #3: Yes

5. Review Comments to the Author

Reviewer #1: This manuscript demonstrated that after oocyte vitrification at the GV stage, increased DSB levels remained in subsequently developing cleavage-stage embryos which was associated with low cell numbers of Day-2 embryos, reduced development to the blastocyst stage and the upregulation of the RAD51 gene at the cleavage stage. Authors concluded that GV-stage oocytes are more susceptible to DNA damages caused by vitrification than pronuclear zygotes. This manuscript has a straightforward experimental structure and provides new evidence for reduced developmental rates due to DNA damage in oocytes during vitrification.

Specific Comments:

L 62; Change " at high rates and after " to " at high rates. After”

L 503; Change " previously [16] which suggests" to " previously [16], which suggests”

Reviewer #2: This manuscript (PONE-D-22-34298) investigated the double-strand breaks (DSBs) in porcine oocytes at GV stage and zygotes after vitrification and their embryos on Day 2, and measured the expression of DNA damage-repair genes in 4-8-cell embryos. The authors found that GV oocytes were more susceptible to DNA damages caused by vitrification than zygotes, which potentially affects their subsequent development to the blastocyst stage. The study is appropriately designed, the methodology is preferable for achieving the specific purposes, and the results are also well presented and discussed. However, some aspects should be improved before accepting this manuscript.

1 Why didn't you investigated the metaphase-MII oocytes?

2 In abstract, “cryoprotectant agents” need not be abbreviated.

3 “metaphase-II” in Line 55 and “GV” in Line 59 should appear in the sentence in Line 50.

4 “in vitro maturation” in Line 62 should be abbreviated.

5 In Line 113-114, is there a difference between without oil coverage and with oil coverage?

6 In Line 147, what are the advantages of modified NCSU-37 used as the basic medium?

7 In Line 154, whether do 60 COCs have the same vitrification effects of 25 COCs?

8 In Line 153, “room temperature” should be abbreviated.

9 The experimental design is too complex, please simplify it.

10 Does the CPA treatment affect the expression of DNA damage-repair genes?

11 What time point did the survival rate of oocytes and zygotes be evaluated?

12 Whether do the vitrification oocytes affect the DSB levels in resultant blastocysts?

Reviewer #3: This is very nice manuscript, well written and presented the novel information of the effects of vitrification of GV oocytes and pronuclear zygotes on developmental potential to blastocysts, DSBs in the DNA, DNA fragmentation, and expression of DNA damage-repair genes.

Only a few comments to authors as follows:

Line 27: ... at the 4-8 cell stage (measured.......)........

Page 5-6: The temperature to culture oocytes, sperm, embryos, have different i.e. 37, 38.5, 39 C. Please explain why did it different?

Line 265: ...... from pooled samples........., please clarify how many embryos pooled?

6. PLOS authors have the option to publish the peer review history of their article (what does this mean?). If published, this will include your full peer review and any attached files.

Reviewer #1: No

Reviewer #2: No

Reviewer #3: No

---

## [Author Response · Author response to Decision Letter 0]

15 Feb 2023

Dear Reviewers,

We would like to thank you for your valuable work and comments which have been very helpful for us to improve the manuscript. The revised version of the manuscript was modified according to your comments. An additional correlation assay was also performed according to the comment of Reviewer 2. We hope that the manuscript now meets the standard of PLOS ONE. The answers to each reviewer comments are listed below.

Answers to Comments

Reviewer 1.

Comment 1: L 62; Change " at high rates and after " to " at high rates. After”

Answer: Corrected as requested.

Comment 2: L 503; Change " previously [16] which suggests" to " previously [16], which suggests” 

Answer: Corrected.

Reviewer 2.

Comment 1: Why didn't you investigated the metaphase-MII oocytes?

Answer: As we emphasized in the introduction (P3L54-59), vitrification of porcine oocytes at the MII stage causes high percentage of fertilization abnormalities (unlike in case of GV-stage vitrification) therefore such oocytes fail to develop to embryos. Even the cleavage percentage is extremely low. Therefore, for MII-vitrified pig oocytes investigations on subsequently developing embryos (experiments 3, 4 and 5 of this study) practically cannot be conducted. Please note that these fertilization abnormalities may not be related to DNA damage since they are caused by cooling-induced premature activation and the damage of the cytoskeleton (spindle) as stated in the introduction (references provided: [9, 10]). For these reasons, MII-vitrified oocytes were not investigated in the current study. We slightly modified the introduction for the better understanding.

Comment 2: In abstract, “cryoprotectant agents” need not be abbreviated.

Answer: Corrected.

Comment 3: “metaphase-II” in Line 55 and “GV” in Line 59 should appear in the sentence in Line 50.

Answer: Corrected as requested.

Comment 4: “in vitro maturation” in Line 62 should be abbreviated.

Answer: Corrected.

Comment 5: In Line 113-114, is there a difference between without oil coverage and with oil coverage?

Answer: Previous studies revealed that oil coverage may exert negative effects on the developmental competence of porcine oocytes/embryos by the adsorption of steroid hormones from media (Shimada et al., 2002 Reproduction 124, 557–564) and as potential source of toxic agents such as peroxides (Martinez et al., 2018 Reprod Dom Anim 53(2):281-286). Indeed batch-dependent toxicity of the paraffin oil did cause some problems for in recent years. However, when oocytes/embryos are cultured in relatively large volumes of medium (e.g., 500 µl or more) under high humidity (approx. 99%), the osmolarity of the medium does not increase to a level which reduces embryo development rates (Martinez et al., 2018 Reprod Dom Anim 53(2):281-286). Accordingly, we did not experience a reduction in blastocyst production efficacy when the embryo culture medium (500 µl) was cultured without oil (data not shown). For these reasons, we avoid oil coverage when oocytes/embryos were cultured in 500 µl media in 4-well dishes (i.e. during IVM and IVC). Previously, we reported high embryo development using this approach (Somfai et al., 2014). In the revised manuscript we added a reference to emphasize this point (P5L114). Furthermore, it is important to point out, that, when culture drops less than 500 µl were used (such as during IVF) we applied oil coverage to prevent osmolarity increase. 

Comment 6: In Line 147, what are the advantages of modified NCSU-37 used as the basic medium?

Answer: The advantages of the modified NCSU-37 medium are as follows; 1) it is completely homemade, which allows the flexible modifications in composition when necessary (which is not possible with TCM199); 2) hence, this variant of NCSU-37 is glucose free to avoid excessive ROS production ( for reference see Karja et al., 2006 Reprod Biol Endocrinol. 4:54. doi: 10.1186/1477-7827-4-54.) and Hepes-buffered for the stable pH outside the incubator; 3) it is simple and cheap (compared with NCSU-23, POM or PXM media). Our previous study demonstrated no harmful effects of treatment of porcine oocytes with the current CPA combination using this medium (Somfai et al., 2013 J Reprod Dev 59(4):378-843). Please note, that the advantages of the modified NCSU-37 as base medium in our protocol are not directly related to the topic of the current study. Therefore, we refrain from discussing it in detail in this study. However, in the revised manuscript we added a reference [21] for the detailed technical description of the vitrification procedure which indeed discusses these points as well. 

Comment 7: In Line 154, whether do 60 COCs have the same vitrification effects of 25 COCs?

Answer: Following the reviewers request, we investigated this issue. Group size does not seem to affect results. We performed correlation analysis on our data (information added to Statistical analysis section). We did not find relationship between the number of oocytes vitrified per group and the percentage of survival neither in GV-vitrified oocytes (R2=0.1908) nor in vitrified zygotes (R2=0.004). We consider this result reasonable since all treatment groups irrespective of size received the exact same treatment (i.e. media, temperature, interval and drop size). Since these results are not directly relevant to the objectives and conclusions, we provide these data as a supplementary file (S2 Fig) and refer to it in the results section (P17L414). We refrain from discussing it to prevent the excessive length of this manuscript. 

Comment 8: In Line 153, “room temperature” should be abbreviated.

Answer: Corrected as requested.

Comment 9: The experimental design is too complex, please simplify it.

Answer: We shortened the experimental design as requested by the reviewer. We believe that the remaining information are essential for the accuracy of the study.

Comment 10: Does the CPA treatment affect the expression of DNA damage-repair genes?

Answer: Experiment 2 demonstrated that CPA treatment did not induce DNA damage in oocytes. Furthermore, our previous studies demonstrated that the current CPA treatment of oocytes did not reduce embryo development. (In the revised manuscript, this point is emphasized in P21L491-494). For these reasons we did not find it reasonable to analyze DNA damage-repair genes in embryos of CPA-treated oocytes.

Comment 11: What time point did the survival rate of oocytes and zygotes be evaluated?

Answer: In both oocytes and zygotes, the survival rate was evaluated after IVF, when the cumulus cells were removed from the zona pellucida as described in P6L134-137 and in experimental design (L310, 322 and 332). 

Comment 12: Whether do the vitrification oocytes affect the DSB levels in resultant blastocysts?

Answer: Thank you for the valuable comment! We agree with the reviewer, it is an interesting question to see if vitrification-related DNA repair processes are still active or already finished in the blastocysts. In fact, we are now investigating this topic in a separate research project. We plan to submit the results separately from this paper to prevent excessive size of this manuscript. We believe that the current results provided are sufficient to support the conclusions of this manuscript.

Reviewer 3.

Comment 1: Line 27: ... at the 4-8 cell stage (measured.......)........

Answer: Corrected.

Comment 2: Page 5-6: The temperature to culture oocytes, sperm, embryos, have different i.e. 37, 38.5, 39 C. Please explain why did it different?

Answer: Thank you for pointing it out! We realized that the temperature (38.5 °C) in L115 was incorrect. We corrected it to 39 °C. In fact, oocyte IVM and embryo culture were both conducted on 39 °C which is the normal body temperature of pigs. On the other hand, sperm preparation, warming and capacitation of frozen sperm was performed at 37 °C a ccording to a previous report (see reference [19] in L124). Normal physiological temperature of male reproductive organs (testicles, epididymis) is lower than the body temperature. Therefore, optimum treatment temperature for sperm during capacitation is generally considered to be approximately 37 °C. 

Comment 3: Line 265: ...... from pooled samples........., please clarify how many embryos pooled?

Answer: As stated in line 268, in experimental each replication 30 embryos were pooled in each group. We modified this sentence for the better understanding.

---

## [Decision Letter · Decision Letter 1]

28 Feb 2023

Vitrification of porcine immature oocytes and zygotes results in different levels of DNA damage which reflects developmental competence to the blastocyst stage

PONE-D-22-34298R1

Dear Dr. Somfai,

We’re pleased to inform you that your manuscript has been judged scientifically suitable for publication and will be formally accepted for publication once it meets all outstanding technical requirements.

Kind regards,

Shao-Chen Sun

Academic Editor

PLOS ONE

Additional Editor Comments (optional):

Reviewers' comments:

Reviewer's Responses to Questions

**Comments to the Author**

1. If the authors have adequately addressed your comments raised in a previous round of review and you feel that this manuscript is now acceptable for publication, you may indicate that here to bypass the “Comments to the Author” section, enter your conflict of interest statement in the “Confidential to Editor” section, and submit your "Accept" recommendation.

Reviewer #2: All comments have been addressed

2. Is the manuscript technically sound, and do the data support the conclusions?

Reviewer #2: Yes

3. Has the statistical analysis been performed appropriately and rigorously? 

Reviewer #2: Yes

4. Have the authors made all data underlying the findings in their manuscript fully available?

Reviewer #2: Yes

5. Is the manuscript presented in an intelligible fashion and written in standard English?

Reviewer #2: Yes

6. Review Comments to the Author

Reviewer #2: The authors have revised the manuscript according to our suggestions and I am glad to agree to accept it

7. PLOS authors have the option to publish the peer review history of their article (what does this mean?). If published, this will include your full peer review and any attached files.

Reviewer #2: No

---

## [Editor Report · Acceptance letter]

9 Mar 2023

PONE-D-22-34298R1 

Vitrification of porcine immature oocytes and zygotes results in different levels of DNA damage which reflects developmental competence to the blastocyst stage 

Dear Dr. Somfai:

I'm pleased to inform you that your manuscript has been deemed suitable for publication in PLOS ONE. Congratulations! Your manuscript is now with our production department. 

Kind regards, 

on behalf of

Prof. Shao-Chen Sun 

Academic Editor

PLOS ONE